# Epstein-Barr Virus-Associated Carcinoma of the Larynx: A Systematic Review with Meta-Analysis

**DOI:** 10.3390/pathogens10111429

**Published:** 2021-11-04

**Authors:** Marcos Antonio Pereira de Lima, Álife Diêgo Lima Silva, Antônio Carlos Silva do Nascimento Filho, Thiago Lima Cordeiro, João Pedro de Souza Bezerra, Maria Aline Barroso Rocha, Sally de França Lacerda Pinheiro, Roberto Flávio Fontenelle Pinheiro Junior, Maria do Socorro Vieira Gadelha, Cláudio Gleidiston Lima da Silva

**Affiliations:** 1School of Medicine, Federal University of Cariri, UFCA, Barbalha 63180-000, Ceará, Brazil; alife.lima@aluno.ufca.edu.br (Á.D.L.S.); antonio.carlos@aluno.ufca.edu.br (A.C.S.d.N.F.); thiago.cordeiro@aluno.ufca.edu.br (T.L.C.); pedro.bezerra@aluno.ufca.edu.br (J.P.d.S.B.); aline.rocha@aluno.ufca.edu.br (M.A.B.R.); sally.lacerda@ufca.edu.br (S.d.F.L.P.); roberto.pinheiro@ufca.edu.br (R.F.F.P.J.); socorro.vieira@ufca.edu.br (M.d.S.V.G.); claudio.gleidiston@ufca.edu.br (C.G.L.d.S.); 2Ceará Cancer Institute, ICC, Fortaleza 60430-230, Ceará, Brazil

**Keywords:** Epstein-Barr virus (EBV), laryngeal carcinoma, carcinogenesis, oncovirus

## Abstract

Over the past few decades, several publications have investigated the role of Epstein-Barr virus (EBV) in head and neck squamous cell carcinomas, and an increasing number of them have shown its presence in laryngeal tumors. The purpose of this meta-analysis was to evaluate the association of EBV with laryngeal carcinoma. The search was carried out in two databases, Scopus and PubMed, using the following terms: “Epstein-Barr virus” and “laryngeal carcinoma”. A total of 187 records were found, of which 31 were selected for meeting the inclusion and exclusion criteria. The meta-analysis yielded an overall pooled prevalence of 43.72% (95% confidence interval (CI): 34.35–53.08). Studies carried out in Europe and Eurasia had slightly higher pooled prevalence than other subgroups, while the prevalence of studies performed in developed countries was higher than in developing countries (46.37% vs. 34.02%). Furthermore, laryngeal carcinoma occurred almost three times as often among EBV-infected individuals compared to those without EBV infection (odds ratio = 2.86 (95% CI: 1.18–6.90); Begg’s test, *p* = 0.843 and Egger’s test, *p* = 0.866). Our findings support the idea that EBV is related to laryngeal carcinoma. However, further studies are needed before recognizing a definitive etiological role of EBV in the development and/or progression of laryngeal carcinomas.

## 1. Introduction

According to the global estimates from the International Agency for Research on Cancer (IARC), for 2018, there were 177,422 new cases and 94,771 deaths due to cancers of the larynx. Both values (incidence and mortality) represented about 1.0% of all 36 cancers analyzed [1]. Histologically, squamous cell carcinoma (SCC) is the most frequent type with 90–95% of the laryngeal cancer cases [2,3]. In turn, laryngeal carcinoma (LC) accounts for 25% to 40% of head and neck malignancies [4].

Although it is well known that tobacco smoking and alcohol consumption are the major risk factors for the development of laryngeal SCC, infectious agents may also be implicated in the pathophysiology of some cases [2,4]. In this scenario, Epstein-Barr virus (EBV) arises as a potential candidate considering all the evidence gathered so far on its oncogenic role in several lymphoid and epithelial malignancies, including head and neck squamous cell carcinomas (HNSCC) [5]. EBV is etiologically linked to undifferentiated nasopharyngeal carcinoma (NPC) [6], and a recent meta-analysis has shown that EBV-infected individuals have a 2.5 times higher risk for developing oral carcinoma [7].

Additionally, various genotypes of human papillomavirus (HPV) have been found in samples of laryngeal SCC [4]. Thus, the laryngeal epithelium seems to share similarities with oral and anogenital epithelia, in which parts of the carcinomas have shown the involvement of the aforementioned viruses [8]. It is worth mentioning that oncoviruses contribute to approximately 12–15% of human cancers worldwide, while EBV and HPV are reportedly involved in 38% of all virus-related cancers [9].

Břicháček et al. [10] were the first to successfully demonstrate the presence of the EBV genome and the latent protein EBNA (Epstein-Barr virus nuclear antigen) in malignant cells of laryngeal carcinomas. The presence of viral DNA in most of the tumor cells suggested that EBV might be associated with LC, resembling EBV-infected NPC. Subsequently, the EBV receptor molecule (CD21) was identified on the surface of primary cultured laryngeal carcinoma epithelial cells [11]. Since then, a few studies have been conducted, applying different techniques such as polymerase chain reaction (PCR), in situ hybridization (ISH), and immunohistochemistry (IHC), and exploring molecular and histological approaches [12,13,14,15,16,17,18,19,20]. Even so, the role of EBV in laryngeal carcinomas is still not fully elucidated. To the best of our knowledge, there is no systematic review, with or without meta-analysis, on this subject in the literature.

In the present study, we performed a systematic review with meta-analysis of articles available in the international literature on the association of EBV with carcinomas of the larynx, in order to answer the following research question: Are EBV-infected individuals at increased risk for the development and/or progression of laryngeal carcinoma? Characteristics of the studies, such as reported prevalence, techniques used, detection targets, and other relevant aspects for understanding of the relationship between EBV and laryngeal carcinomas, were evaluated.

## 2. Results

### 2.1. Characteristics of the Studies

The 31 systematically reviewed studies were carried out in 19 different countries in Europe (n = 12) [6,10,11,12,13,14,15,16,17,18,19,20], Asia (*n* = 8) [21,22,23,24,25,26,27,28], North America (*n* = 5) [3,29,30,31,32], Eurasia (*n* = 3) [4,5,33], Oceania (*n* = 1) [34], South America (*n* = 1) [2], and concomitantly in North America and Europe (*n* = 1) [35] (Table 1).

The studies have detected EBV by PCR-based techniques (*n* = 16) [3,4,5,12,13,15,17,18,21,23,25,28,31,32,33,34], in situ hybridization (ISH) (*n* = 14) [2,10,12,14,16,19,20,22,26,27,28,31,32,35], or immunological methods (*n* = 10) [5,10,12,14,16,24,28,29,30,35]. In seven studies, two techniques were used for viral detection, with five studies combining immunological assays and hybridization [4,10,14,16,35]: a study using immunological and PCR-based techniques [5] and another study applying ISH and PCR-based techniques [32]. Three different techniques (ISH, immunological, and PCR-based methods) were employed in two studies [12,28].

Some articles also explored the presence of other viruses. In order to investigate the possible association between viral (co)infection and a laryngeal carcinogenesis, these data were also considered in the present systematic review and summarized in Table 1. HPV has been investigated in 12 studies [2,12,15,16,18,19,20,23,25,26,34,35]. A few studies assessed other human herpesvirus (HHV) members, namely, human herpesvirus type 1 (HHV-1, also known as herpes simplex virus type 1, HSV-1) was investigated in three studies [17,29,35]; HHV-2 (or HSV-2) in three studies [10,29,35]; cytomegalovirus (CMV or HHV-5) in one study [17]; and HHV-8 (also known as Kaposi’s sarcoma herpesvirus—KSHV) in two studies [16,35]. The presence of human immunodeficiency virus (HIV) was explored in two studies [16,35]. Some human polyomaviruses were also analyzed in four studies, including Merkel cell polyomavirus (MCV or MCPyV) in two studies [3,20]; BK virus (BKV) in one [18]; and the viral load of John Cunningham virus (JCV) in another one [25].

### 2.2. Meta-Analysis

Collectively, the studies included in the present review gathered 1292 distinct specimens from patients with laryngeal carcinoma. Twenty-nine studies could be submitted to meta-analysis, of which eight studies presented a prevalence of 0 or 100% and were automatically excluded from the analysis. The overall pooled prevalence was 43.72% (95%CI: 34.35–53.08; *p* < 0.001). Figure 1 shows the forest plot, applying the random-effects model due to the observed heterogeneity, with the prevalence of EBV and respective 95% CI for each study presented by subgroups established based on the type of technique used (immunological, in situ hybridization, or PCR-based method). The highest pooled prevalence was observed in the subgroup that used immunological techniques for EBV detection in LC: 50.68% (95% CI: 28.58–72.80; *p* < 0.001). Subgroups that used the in situ hybridization and PCR-based technique yielded the following pooled prevalence: 49.58% (95% CI: 28.47–70.68; *p* = 0.004) and 39.31% (95% CI: 27.84–50.77; *p* < 0.001), respectively.

Table 2 displays the results of the meta-analysis of the studies using the random-effects model, stratified by sampling method, world region, and degree of development of the country where the samples were collected. The pooled prevalence was similar in the following subgroups: serum; fresh frozen tissue; and formalin fixed, paraffin embedded tissues (FFPET). The highest pooled prevalence was observed in a subgroup consisting of a single study that used throat washings as samples; for this reason, no *p*-value was presented by the software (64.51%; 95% CI: 47.67–81.35).

Studies carried out in Europe and Eurasia had slightly higher pooled prevalence than other subgroups, whereas the prevalence of studies performed in developed countries (46.37%; 95% CI: 35.27–57.47; *p* < 0.001) was higher than in developing countries.

In our sample, 15 case-control studies were identified and used in the odds ratio analysis [4,5,6,12,13,21,22,24,25,27,28,29,30,33,35]. Based on the results of the heterogeneity tests, the random-effects model was used to determine the odds ratio (Figure 2). The analysis between the LC versus normal control group revealed that EBV-infected individuals are almost three times more likely to develop laryngeal carcinoma (OR = 2.86 (95% CI: 1.18–6.90); *p* = 0.002); in comparison, for the control group composed of non-malignant lesions, the chance of developing LC in EBV-positive individuals was nearly twice as common as in the non-infected group (OR = 1.94 (1.18–3.21); *p* = 0.299). The results of the *p*-value and I-squared statistics in the latter analysis revealed low heterogeneity (I^2^ < 50% and *p* > 0.05), therefore indicating the fixed-effects model as the most suitable, and both models had similar odds ratio results (OR = 2.00; 95% CI: 1.30–3.06; fixed-effects model). The overall result showed values closely related to those of the second subgroup (OR = 2.18 (95% CI: 1.33–3.59); *p* = 0.007). Figure 3 presents a funnel plot along with results of two statistical tests to assess the risk of publication bias: Begg’s test (*p* = 0.843) and Egger’s test (*p* = 0.866). No evidence of publication bias was observed.

## 3. Discussion

### 3.1. EBV and Laryngeal Carcinoma

The hypothesis that viral infections are associated with the development of human cancers has been studied since the early 1960s, when the first human oncovirus was discovered in Burkitt’s lymphoma: the Epstein-Barr virus [36]. In 1973, Wolf, zur Hausen, and Becker [37] demonstrated by nucleic acid hybridization that epithelial cells of nasopharyngeal carcinomas harbor the EBV genome, which represented a breakthrough in this field by showing, for the first time, the involvement of EBV in epithelial malignancies.

One year later, the first study evaluating the involvement of EBV in laryngeal carcinoma was published; however, the authors failed to show any significant antibody titer differences between carcinoma and control groups, leading them to suggest that EBV is not etiologically related to LC [29]. Perhaps for this reason, there was a gap of nearly a decade until the next study investigating EBV in LC. Unlike the former, in the study by Břicháček et al. [10], the presence of EBNA (Epstein-Barr virus nuclear antigen) and EBV DNA was revealed in tumor cells of three patients with supraglottic laryngeal carcinoma (SGLC). Furthermore, the viral DNA was present in most if not all the tumor cells and, in two out of three EBV-infected tumors, foci of well-differentiated cells with keratinization were observed, indicating that EBV may be associated not only with undifferentiated carcinoma, a common feature of EBV-positive nasopharyngeal carcinomas. Although the authors suggest that at least some of these tumors are associated with EBV, they also recognize that these findings do not necessarily imply that EBV is etiologically involved in SGLC.

In the same year, Callaghan et al. [30] found the geometric mean titers (GMT) of IgG anti-VCA (viral capsid antigen) to be tenfold higher in patients with cancer of the larynx when compared to controls. According to the authors, the elevated IgG anti-VCA titers may indicate an activation of a latent EBV infection occurring after a state of immunosuppression which occurs in the host due to the effects of the tumor. Additionally, the GMTs for IgA anti-VCA in LC were significantly elevated compared to those of the control group. On the other hand, in the study by Lee et al. [28], serum IgA anti-VCA positivity was not associated with laryngeal EBV DNA positivity, but instead indicated previous repeated EBV infections or frequent reactivation of latent EBV in B lymphocytes. In 1989, a work identified a 200 kDa protein on the surface of primary cultures of a laryngeal carcinoma sharing an epitope with the C3d/EBV receptor molecule CD21 of B-cells [11].

Thereafter, other studies have sought to assess the relationship between EBV and laryngeal carcinoma, and the number of publications on this subject gradually increased in the following decades, but in a smaller proportion compared to other EBV-related tumors (such as nasopharyngeal and gastric carcinomas). As shown in Table 1, the prevalence of EBV in laryngeal carcinoma demonstrated variable results, ranging from 0 to 100%. It is worth noting that the two studies which presented prevalence of 100% applied serological assays. Moreover, the sample sizes, techniques employed, and targets also varied among the publications which, according to Goldenberg et al. [32], are partially responsible for the contradictory findings.

### 3.2. Controversial Aspects of the EBV–LC Relationship

Among the studies included, some were not able to detect EBV in their samples, either by ISH [2,16,19,20,22,26,31], immunohistochemistry (IHC) [16], or by PCR–Southern blot [34]. Furthermore, the studies by Lewensohn-Fuchs et al. [12] and Goldenberg et al. [32] detected EBV using nested-PCR and qPCR, respectively, but could not confirm viral presence by IHC and/or ISH.

It is worth mentioning that studies such as Yang et al. [22] and de Oliveira et al. [2] did not find ISH-positive cases even when evaluating expressive samples sizes: 93 and 110, respectively. These results are consistent with most studies involving EBV, in which oropharyngeal, hypopharyngeal and laryngeal carcinomas rarely show signals for this virus [22].

Regarding the studies employing PCR-based methods, most demonstrated positivity for EBV with an average prevalence around 50%. However, these results face the same dilemma observed in carcinomas from other anatomical sites, that is, the real meaning of the expressive PCR positivities given the impossibility of determining the precise source of viral DNA, whether from tumor or non-malignant cells (including lymphocytes and non-neoplastic epithelial cells). In this regard, when PCR-based methods are used for EBV detection, the results must be considered with caution because it has been estimated that EBV latently infected B-cells may be found in circulating cells in healthy individuals, and thus false-positive results must be considered [2,3]. In a study using qPCR, the authors assumed that the low levels of EBV detected by this highly sensitive molecular method may be related to the presence of the EBV genome in rare lymphoid or epithelial cells adjacent to the primary cancer [32].

In the present review, even without applying a time limitation filter in the searches, the number of articles retrieved evaluating the involvement of EBV in LC is still small. Moreover, many studies enrolled small samples, some applied indirect EBV detection methods (e.g., serological assays), while only a few were able to demonstrate the viral genome or its products (transcripts or proteins) in tumor cells, and there is a lack of information on the distribution patterns of staining. In addition, there are some relevant confounding variables, such as alcohol consumption and tobacco smoking that are traditional risk factors for HNSCC, as well as other oncogenic viruses that are also found in this site, sometimes in coinfections, which compromise the assessment of the role of EBV in the initiation and/or progression of laryngeal carcinomas. In this context, Mulder et al. [20] were unable to detect EBV in nine LC cases of non-smokers and non-drinkers.

Therefore, some authors recognize that the role of EBV in the pathogenesis of laryngeal carcinoma is still unclear [2,3,4,19,20,22,25,32,33]. In this context, Muderris et al. [4] assert that EBV is a very common virus which can be found in the mucosal cells of the upper aerodigestive tract in a considerable proportion of the population, and although EBV is present in cancer tissues of some of the LC cases, its presence has no effect on the pathogenesis of laryngeal carcinomas.

In a Turkish study, the identification of EBV DNA in 50% of patients with laryngeal carcinoma and in 41.2% of the vocal cord nodules indicates, according to the authors, that this agent does not seem to be directly associated with the pathogenesis of LC, but suggests a causal relationship involving EBV and proliferative diseases in the larynx [33]. In the study by Abdulamir et al. [24], serum levels of EBV IgG and IgA antibodies in CL patients were not significantly higher than those in the control group.

Three studies could not find any significant difference between EBV DNA positivity and smoking habits [4,32,33]. Two found no significant differences between EBV positivity and alcohol consumption and tumor grade [4,32], while two studies found no significant differences between EBV positivity and tumor stage [13,33]. Kiaris et al. [13] found no association between the presence of EBV with tumor differentiation and lymph nodes metastasis (N). A Greek study demonstrated that the EBV has no impact on tumor recurrence and the survival of the patients [15].

Interestingly, two studies failed to detect EBV by EBER-ISH in laryngeal lymphoepithelial carcinomas, denoting the peculiarity of this rare type of neoplasm [19,31]. Acuña et al. [19] estimated an annual incidence of 0.013/100,000 and prevalence in 0.2% of laryngeal and hypopharyngeal lymphoepithelial carcinomas. The origin of this kind of carcinoma of the larynx remains controversial. Only a third of the cases registered in the literature have shown the presence of EBV. It is a much smaller number when compared to the participation of this virus in nasopharyngeal carcinomas, where almost all cases are related to EBV [19]. Furthermore, the results of the aforementioned studies indicate that the percentage of involvement of EBV in these tumors appears to be even lower.

### 3.3. Evidence That Corroborates the Role of EBV in the Development of LC

Despite the controversy, some authors consider that EBV plays a role in the development of LC [5,6,13,21,23,25,27]. This meta-analysis revealed a relevant EBV pooled prevalence of 43.72% (Figure 1), and that EBV-infected individuals are 2.86 times more likely to develop laryngeal carcinoma when compared to the normal control (Figure 2). In the analysis with the control group composed of non-malignant lesions, the result of the odds ratio was 1.94 (95% CI: 1.18 to 3.21), while the overall analysis showed that EBV-infected individuals are at a 2.18 times greater risk of developing laryngeal carcinoma.

Although the possibility of false-negative cases is admitted in those studies investigating the presence of EBV in samples taken from FFPET, since formalin is a known inhibitor for PCR [4,27], in our analysis of subgroups, the employment of FFPET showed pooled prevalence at the same level as the other subgroups (serum and fresh frozen tissue) (Table 2). However, when separated into subgroups based on the type of technique, the PCR-based methods had the lowest pooled prevalence of 39.31% (Figure 1), and, indeed, more than half of the studies in this subgroup used FFPET. In addition, there were no expressive differences in the pooled prevalence by world regions.

Interestingly, the developed countries demonstrated higher pooled prevalence than developing ones. Although there is little information in this regard, one could argue that the difference observed between developing and developed countries might be related to the period of primary EBV infection. According to Ali et al. [27], due to some cultural practices, EBV exposure usually occurs in early childhood in developing countries, and primary infection in young children is typically associated with an unremarkable acute syndrome, while infection is often delayed in developed countries, and acute primary infection occurring in adolescence or adulthood can result in a self-limiting lymphoproliferative disorder known as infectious mononucleosis (IM). However, there is insufficient evidence to affirm that the delayed primo-infection may be related to the increased pooled prevalence verified in developed countries. Also, the difference in pooled prevalence may be due to the number of studies from developed and developing countries, 22 versus 6 (Table 2); nevertheless, the sample sizes were not that different: 692 versus 600.

In a recent case-control study, the EBV DNA positivity detected by qPCR was significantly higher in LC group (52% vs. 20%), showing that individuals with laryngeal EBV DNA positivity had a greater risk of developing LC (unadjusted OR = 4.3). Also, EBV DNA presence was positively correlated with BCL-2 expression (an anti-apoptotic protein) [28]. Although most EBER-ISH signals were localized into the nuclei of tumor-infiltrating lymphocytes, which according to the authors suggests the occurrence of the hit-and-run mechanism of carcinogenesis at this epithelium, the high laryngeal EBER signal was identified as a poorer prognostic factor in LC.

Despite the debate on the high sensitivity of PCR-based methods which may incur false-positive results, in a multivariate analysis performed by Lee et al. [28], EBV DNA positivity was identified as one of the independent risk factors for LC (OR = 39.7; 95% CI: 3.3–478.0; *p* = 0.004).

Regarding histological approaches, although evidence on the location of EBV is scarce, some studies have shown distinct nuclear staining by ISH in tumor cells of laryngeal SCC [10,28,35]. It is worth mentioning that Břicháček et al. [10], unlike the others, employed ISH targeting the viral genome and were able to detect EBV in almost all tumor cells.

The cases in which EBV DNA was found but viral protein was not detected [12,32] may indicate that the virus is present but transcriptionally inactive [13]. In this scenario, other methods, such as the detection of EBV-RNA transcripts in tissues, can provide a broader understanding of transcriptional activity in latent and lytic EBV infection, which could bring new insights on its pathogenic role [3].

Some experiments have indicated that EBV infection may affect the immune response in the context of laryngeal carcinogenesis. For instance, the findings of Klatka et al. [6] point to an increased early activation of T-lymphocytes with reduction of the CD25+ population during recent or recurrent EBV infection and suggest that dysfunction of immune response in patients with LC might be associated with EBV infection [6]. In another study, the EBV infection seemed to have a definite effect in the secretion of immunosuppressive cytokines such as interleukin 10 (IL-10) and transforming growth factor β1 (TGFβ1). This suggests that cytokines may play a pathogenic role in EBV-associated LC, by both suppressing anti-tumor responses and by promoting tumor growth [5].

### 3.4. The Role of the Coinfections

The importance of studies focusing on the impact of coinfections on the malignant transformation of the laryngeal epithelium have also been highlight, which may contribute to clarifying their association with the carcinogenic process and to guide prophylactic strategies [3,17,18]. In the opinion of Drop et al. [18], pathogenic infection may contribute but is not sufficient for the oncogenic process; therefore, a secondary coinfection with another virus may act as an important cofactor in triggering oncogenesis and/or tumor progression. The interaction of multiple oncoviruses may be an important risk factor in the neoplastic development that must be considered [3]. In a study utilizing nested-PCR, 15% (6/40) of the LC cases showed HPV/EBV coinfection [18]. In this sense, Liu et al. [23] suggest an association of laryngeal carcinogenesis and infection with the high-risk HPV-16, HPV-18, and EBV.

In our analysis, five studies reported EBV and HPV coinfections [3,12,15,17,18]. The present findings denote that laryngeal mucosa is another site that can harbor both agents simultaneously, likewise other HNSCC, as well as anogenital and breast carcinomas. Furthermore, it suggests that epithelium of the larynx might be susceptible to the same pattern of viral cooperation verified in the aforementioned carcinomas. The meaning of the coexistence of HPV and EBV in oral and anogenital carcinomas has already been addressed by our team in a previous review [8].

Although the role of EBV/HPV coinfection has been hypothesized, the coexistence of both viruses has not been proven to have an unfavorable implication in the prognosis of LC patients [15]. In addition, the results obtained by de Oliveira et al. [2] corroborate the hypothesis that HPV infection, but not EBV infection, has a role in the pathogenesis of a subset of laryngeal carcinomas. In fact, as shown in Table 1, some studies have evaluated the prevalence of coinfections between HPV and other viruses [3].

Other viruses detected in LC include HSV-1, CMV, HHV-8, HIV, BKV, and MCPV [3,10,17,18,35]. Additionally, Zheng et al. [25] detected a slightly higher viral load level of JCV in LC cases compared to laryngeal normal mucosa, but without statistical significance. Drop et al. [18] reported HPV + EBV + BKV coinfection in 4 out of 40 (10%) LC cases. However, there are insufficient reports describing the epidemiology of viral coinfections in laryngeal carcinomas; hence, the role of these viruses in the pathogenesis of LCs is not yet fully elucidated [3,5].

In general, the limitations of the present study were related to the use of only two databases and the inclusion of articles exclusively published in English that may have restricted the number of records found. Moreover, as studies on EBV in laryngeal carcinoma are less common, a time period was not established in the searches, resulting in the selection of a greater number of articles using indirect methods for EBV detection, which along with the small sample sizes observed in some works and diversity of techniques employed are likely the main causes of the heterogeneity verified in our meta-analysis. Finally, the review being centered on observational studies, due the nature of the investigated subject, may have downgraded the strength of the current evidence.

## 4. Materials and Methods

### 4.1. Search Strategy

To investigate the association of EBV with laryngeal carcinoma, we conducted a systematic review with a meta-analysis of publications that were retrieved from the electronic scientific databases PubMed and Scopus, using the following search terms: “Epstein-Barr virus” and “laryngeal carcinoma”. In the PubMed database, the search was set to include all fields, while in Scopus the search was conducted based on the title, abstract, and keywords. In view of the limited number of articles available, no filter was applied for the time period; for this reason, the search covered a relatively long period (November 1973 to 3 September 2021).

The manuscripts were selected primarily through the analysis of titles and abstracts. The publications were reviewed independently by three authors and submitted to the following inclusion criteria: (i) articles written in English; (ii) articles addressing the association between carcinoma of the larynx and EBV; and (iii) original studies. Publications were excluded which (i) employed no EBV detection method (direct or indirect); (ii) were not classified as original research (i.e., letters to the editor, prefaces, comments, editorials, reviews, book chapters, and case reports); (iii) involved only squamous cell carcinomas from other locations in the upper aerodigestive tract and did not include any laryngeal cases, or combined the LC cases with tumors from other anatomical sites without presenting the values separately; (iv) were based on animal samples or models; or (v) were repeated documents. Any disagreement during the selection of records was resolved by the senior author.

Initially, we found 109 records in PubMed and 78 in Scopus. Fifty-nine duplicate records were excluded. After filtering for language and type of study and evaluating the connection of the content with the proposed aim, 31 articles were included in the systematic review (Figure 4). Full-text articles were obtained through the website of Coordination for the Improvement of Higher Education Personnel (CAPES), a restricted-access online library maintained by the Brazilian Ministry of Education. This review followed the Preferred Reporting Items for Systematic Reviews and Meta-Analyses (PRISMA 2020) protocol. The protocol/review was not registered in any database. Ethics committee approval was not required.

### 4.2. Statistical Analysis

The statistical analysis was performed with the software Stata, version 12.0 (StataCorp; College Station, TX, USA), using the following commands: “metan”, “metafunnel”, and “metabias” [38]. First, EBV prevalence and confidence intervals (CI) were calculated for each study. The 95% CI was calculated using the Wilson method [39]. Then, we performed a meta-analysis to estimate the overall pooled prevalence of EBV. Analyses of subgroups were also performed after stratifying the potential sources of heterogeneity among the studies, including (i) viral detection technique (hybridization, immunological, or PCR-based methods), (ii) specimen collection method, (iii) geographic location of the study, and (iv) country’s development status. Articles lacking detailed information (total number of cases and/or positive cases) for each type of specimen were excluded from the analysis due to the impossibility of calculating confidence intervals. The odds ratio (OR) and the corresponding 95% CI were estimated using the case-control studies. The heterogeneity among the results of the studies was evaluated with the I-squared (I^2^) and the *Q* statistics, with the level of statistical significance set at 5% (*p* < 0.05). The EBV pooled prevalence was initially estimated with a fixed-effects model. However, due to the significant level of heterogeneity observed, we conducted a second analysis using the random-effects model of Der Simonian and Laird [40]. Studies reporting a prevalence of 0% or 100% were automatically excluded from the analysis. To assess publication bias, we used funnel plot and statistical tests (Egger’s test and Begg’s test).

## 5. Conclusions

To our knowledge, the present study represents the first systematic review with meta-analysis of the literature on the association of EBV with carcinoma of the larynx, with relevant evidence such as the expressive pooled EBV prevalence of 43.7% in samples from LC patients and the fact that EBV-infected individuals are 2.86 times more likely to develop laryngeal carcinoma. Despite the controversies, our findings indicate a possible role for EBV as a risk (co)factor in the development and/or progression of LC. Moreover, these tumors seem to share similarities with other HNSCCs, as well as with anogenital and breast carcinomas in which EBV can be found alone or in coinfection with high-risk genotypes of HPV, strengthening the thesis that both oncoviruses can cooperate in the epithelial transformation. There may also be an influence of EBV in the immunological response in the context of laryngeal carcinoma, providing an appropriate environment for the establishment of the LC, by promoting tumor growth and by suppressing anti-tumor responses. On the other hand, unlike other EBV-related carcinomas, there are still few studies in the literature dealing with this topic in the larynx; consequently, the location and distribution of EBV in the tissue remain controversial and there is a lack of information about its transcriptional activity, and, in turn, about the expression pattern of viral genes, whose available data only allow us to assume that EBERs are expressed in these tumors. Hence, further studies are needed in order to elucidate the inconsistencies. In this sense, large and standardized case-control and cohort prospective studies, focusing on cases unrelated to confounding factors for the upper aerodigestive tract, combining different techniques and targets for EBV detection, may contribute to the understanding of the role played in this context. Also, in vitro studies are particularly important to investigate the effects of EBV on laryngeal carcinoma cell lines and possible underlying tumor mechanisms. Taken together, the present findings suggest that EBV may be associated with laryngeal carcinomas and contribute to expanding the knowledge on this oncovirus, in addition to encouraging new studies which, ultimately, may improve the clinical management and prevention of LC cases.

## Figures and Tables

**Figure 1 pathogens-10-01429-f001:**
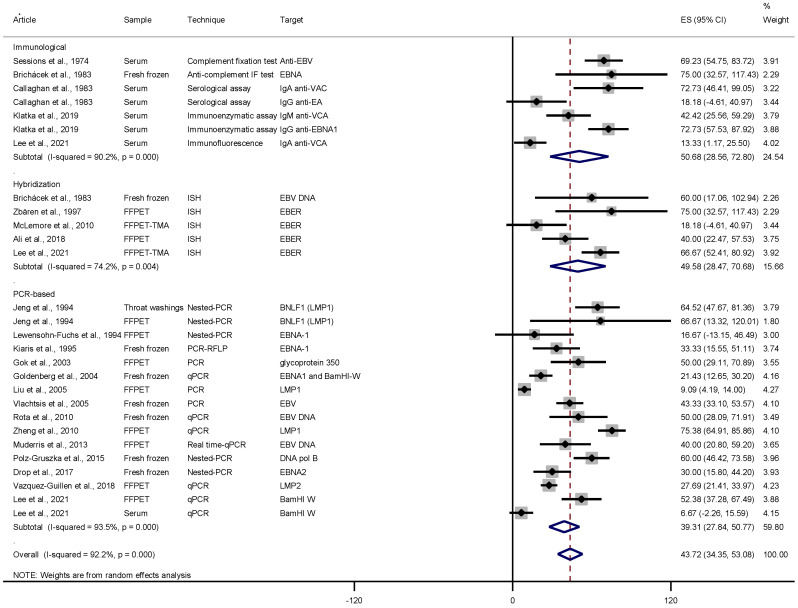
Pooled prevalence of EBV stratified by techniques (ISH, immunological, and PCR-based methods). Dashed lines indicate mean prevalence.

**Figure 2 pathogens-10-01429-f002:**
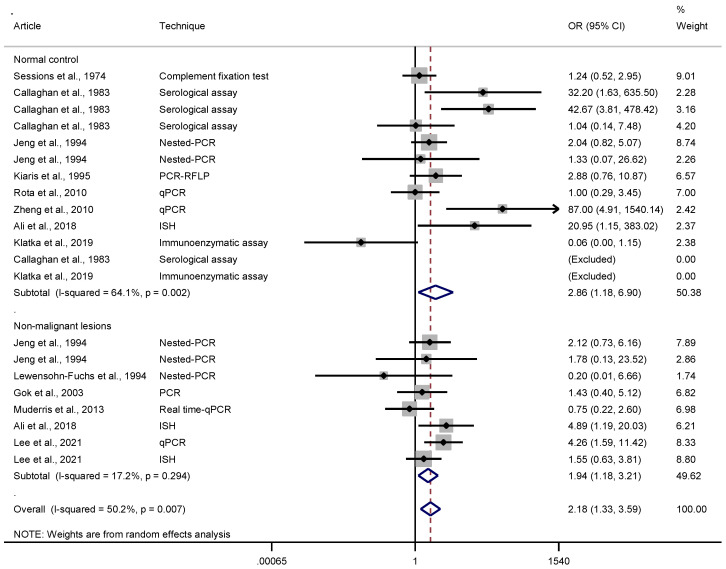
Odds ratio (OR) of laryngeal tissues associated with EBV (cases vs. controls), using a random-effects model, stratified by type of control (normal control and non-malignant lesions).

**Figure 3 pathogens-10-01429-f003:**
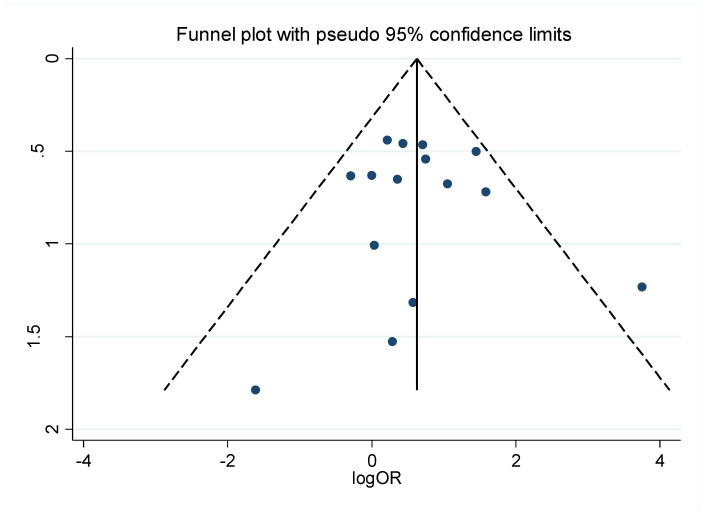
Funnel plot and respective publication bias statistics for the case-control studies (Begg’s test: *p* = 0.843; Egger’s test: *p* = 0.866).

**Figure 4 pathogens-10-01429-f004:**
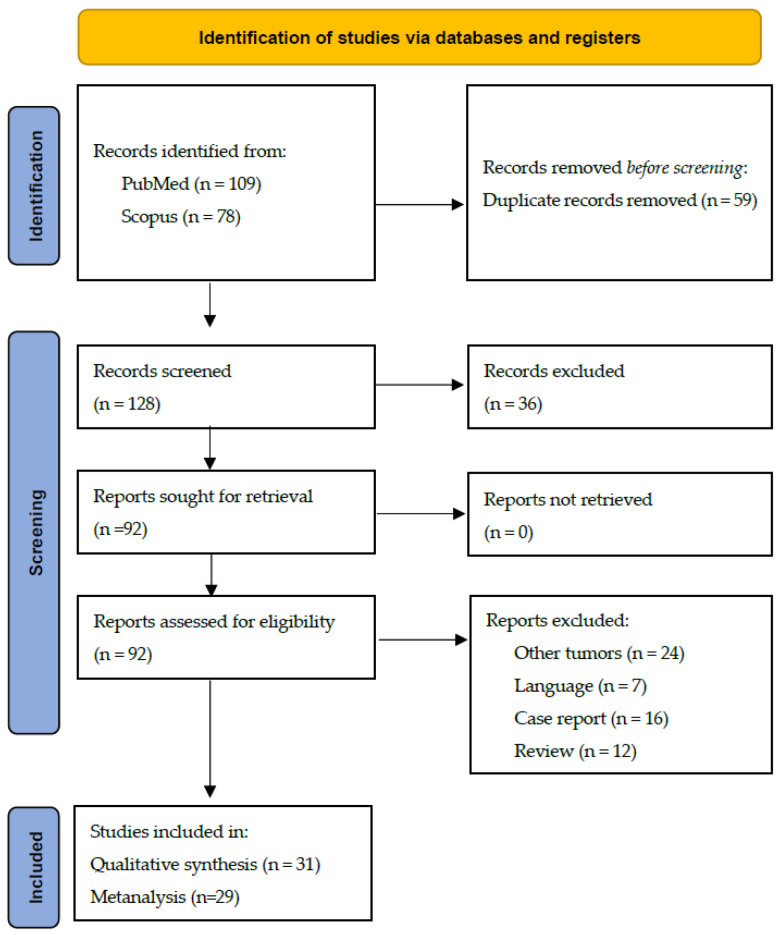
Flow diagram synthesizing the selection procedure of studies for the review. PubMed and Scopus databases.

**Table 1 pathogens-10-01429-t001:** Summary of the 31 studies that evaluated EBV in laryngeal carcinomas.

Country	Specimen	Age Range		EBV			EBV				HPV		Co-Infection	Other Viruses	Authors	Year
				(Carcinoma)			(Control Group)									
			Techniques	Target	Prevalence	Sample Control	Techniques	Target	Prevalence	Technique and Target	Prevalence	Genotypes Identified	HPV/EBV			
USA	Serum	N.A.	Complement fixation test	Anti-EBV	27/39 (69.2%)	Serum	Complement fixation test	Anti-EBV	38/59 (64.4%)	-	-	-	-	HHV-1 40/40 (100%) /HHV-2 39/39 (100%)	Sessions et al. [29]	1974
Czech Republic	Fresh frozen	From 46 to 69 years old	ISH	EBV DNA	3/5 (60%)	-	-	-	-	-	-	-	-	HSV 1/5 (20%)	Brichácek et al. [10]	1983
			Anti-complement immunofluorescence test	EBNA	3/4 (75%)	-	-	-	-	-	-	-	-	-		
			Serology	IgG anti-VAC	Mean antibody titers/5 (72%)	-	-	-	-	-	-	-	-	-		
				IgA anti-VAC	Mean antibody titers/5 (22%)	-	-	-	-	-	-	-	-	-		
				IgG anti-EA	Mean antibody titers/5 (32%)	-	-	-	-	-	-	-	-	-		
				IgA anti-EA	Mean antibody titers/5 (12%)	-	-	-	-	-	-	-	-	-		
USA	Serum	N.P.	Serological assay	Anti-EBNA	GMT = 54.8	Healthy control	Serological assay	EBNA	GMT = 13.8	-	-	-	-	-	Callanghan et al. [30]	1983
				IgG anti-VAC	11/11 (100%)			IgG anti-VAC	7/17 (41.2%)	-	-	-	-	-		
				IgA anti-VAC	8/11 (72%)			IgA anti-VAC	1/17 (5.9%)	-	-	-	-	-		
				IgG anti-EA	2/11 (18%)			IgG anti-EA	3/17 (17.6%)	-	-	-	-	-		
				IgA anti-EA	0/11 (0%)			IgA anti-EA	0/17 (0%)	-	-	-	-	-		
Taiwan	Throat washings	N.A.	Nested-PCR	BNLF1 (LMP1)	20/31 (64.5%)	Nonmalignant diseases	Nested-PCR	BNLF1 (LMP1)	12/26 (46.2%)	-	-	-	-	-	Jeng et al. [21].	1994
						Healthy control	Nested-PCR	BNLF1 (LMP1)	25/53 (47.2%)	-	-	-	-	-		
	FFPET		Nested-PCR	BNLF1 (LMP1)	2/3 (66.6%)	Nonmalignant diseases	Nested-PCR	BNLF1 (LMP1)	9/17 (52.9%)	-	-	-	-	-		
						Healthy control	Nested-PCR	BNLF1 (LMP1)	3/5 (60%)	-	-	-	-	-		
Sweden	FFPET	N.A.	Nested-PCR	EBNA-1	1/6 (16.6)	Laryngeal papilloma	Nested-PCR	EBNA-1	1/2 (50%)	Nested-PCR and E5 region	1/6 (16.6%)	HPV 31	1/6 (16.6%)	-	Lewensohn-Fuchs et al. [12]	1994
			ISH	EBERs	0/1 (0%)	-	-	-	-	-	-	-	-	-		
			IHC	BZLF1	0/1 (0%)	-	-	-	-	-	-	-	-	-		
Greece	Fresh frozen	N.A.	PCR-RFLP	EBNA-1	9/27 (33.3%)	Adjacent normal tissue	PCR-RFLP	EBNA-1	4/27 (14.8%)	-	-	-	-	-	Kiaris et al. [13]	1995
USA	FFPET	From 51 to 82 years old Mean: 64	ISH	EBER-1	0/4 (0%)	-	-	-	-	-	-	-	-	-	MacMillan et al. [31]	1996
Switzerland	FFPET	From 53 to 73 years old Mean: 65.75	ISH	EBER	3/4 (75%)	-	-	-	-	-	-	-	-	-	Zbären et al. [14]	1997
		Median: 68.5	IHC	LMP1	0/4 (0%)	-	-	-	-	-	-	-	-	-		
South Korea	FFPET-tissue microarray	N.P.	ISH	EBERs	0/93 (0%)	-	-	-	-	-	-	-	-	-	Yang et al. [22]	2001
Turkey	FFPET	Mean: 52	PCR	gp350	11/22 (50%)	Vocal cord nodules (Mean 38 years)	PCR	gp350	7/17 (41.17%)	-	-	-	-	-	Gök et al. [33]	2003
USA	Fresh Frozen	N.A.	qPCR	EBNA1 and BamHI-W	18/84 (21.42%)	-	-	-	-	-	-	-	-	-	Goldenberg et al. [32]	2004
	FFPET		ISH	EBER	0/2 (0%)	-	-	-	-	-	-	-	-	-		
China	FFPET	From 38 to 75 years old	PCR	LMP1	12/132 (9.13%)	-	-	-	-	PCR and E6 region	68/132 (52.03%)	HPV16	-	-	Liu et al. [23]	2005
		Mean: 57.32				-	-	-	-	PCR and E6 region	40/132 (30.89%)	HPV18	-	-		
						-	-	-	-	PCR and L1 gene	9/132 (7.3%)		-	-		
Greece	Fresh frozen	From 40 to 77 years old Median: 62	PCR	EBV	39/90 (43.3%)	-	-	-	-	PCR HPV and 16 and 18 DNA	36/90 (40%)	HPV16 and HPV 18	19/90 (21.1%)	-	Vlachtsis et al. [15]	2005
Brazil	FFPET	From 25 to 86 years old Mean: 59.4	ISH	EBER1	0/110 (0%)	-	-	-	-	Multiplex PCR and L1 gene	41/110 (37.3%)	HPV16 and HPV 18	-	-	De Oliveira et al. [2]	2006
Australia	FFPET	From 12 to 15 years old Mean 13.67 Median 14	PCR–Southern blot	BamHI-W	0/3 (0%)	-	-	-	-	PCR-ELISA and L1 gene	2/3 (66.6%)	HPV16	-	-	Chow et al. [34]	2007
Iraq and Jordan	Serum	Median: 50.5 Mean: 50.81	ELISA	IgA anti-VCA	0.1 ± 0.04 (66 cases of LC)	Serum (patient free of cancer or any other medical illness)	ELISA	IgA anti-VCA	Mean ELISA OD EBV serum/100 (0.094 ± 0.007)	-	-	-	-	-	Abdulamir et al. [24]	2008
				IgG anti-VCA	0.132 ± 0.009	(age- and sex-matched control subjects)	ELISA	IgG anti-VCA	Mean ELISA OD EBV serum (0.13 ± 0.018)	-	-	-	-	-		
Spain	FFPET	From 36 to 54 years old	ISH	EBER1 and 2	0/6 (0%)	-	-	-	-	PCR and HPV DNA	0/6 (0%)		-	All cases HIV(+)	Moyano et al. [16]	2009
		Mean: 42.6 Median: 41.5	IHC	LMP1	0/6 (0%)	-	-	-	-				-	HHV-8 0/6 (0%)		
Turkey	Fresh frozen	Mean: 54.6	qPCR	EBV DNA	10/20 (50%)	Patients with acute EBV infection	qPCR	EBV DNA	10/10 (100%)	-	-	-	-	-	Rota et al. [5]	2010
						Healthy individuals without EBV infection	qPCR	EBV DNA	0/10 (0%)	-	-	-	-	-		
Japan	FFPET	From 27 to 91 years old Mean: 65.5	qPCR	LMP1	49/65 (75.3%)	Non-neoplastic mucosa	qPCR	LMP1	0/14 (0%)	qPCR and HPV16 and 18 DNA	-	HPV 16	-	JCV (prevalence not found)	Zheng et al. [25]	2010
USA and Spain	FFPET-tissue microarray	From 33 to 61 years old Mean: 47.06 Median: 46.5	ISH	EBER	2/11 (18.2%)	-	-	-	-	PCR dot blot-hybridization and HPV DNA	0/13 (0%)	25 types of HR-HPV (especially HPV type 16)	-	HIV 5/15, (33.3%), HSV-1 2/7, (28.6%), HSV-2 0/6 (0%), HHV-8 1/7 (14.3%)	McLemore et al. [35]	2010
			IHC	LMP1	0/6 (0%)	-	-	-	-				-			
South Korea	FFPET-tissue microarray	N.A.	ISH	EBER	0/29 (0%)	-	-	-	-	-	-	-	-		Choe et al. [26]	2012
Turkey	FFPET	From 42 to 67 years old Mean: 54.6	Real time-qPCR	EBV DNA	10/25 (40%)	Benign laryngeal lesion (mean: 48.8 years)	Real time-qPCR	EBV DNA	8/17 (47.1%)	-	-	-	-		Muderris et al. [4]	2013
Poland	Fresh frozen	N.P.	Nested-PCR	DNA pol B	30/50 (60%)	-	-	-	-	PCR L1 region	18/50 (36%)	-	(15%)	CMV (4/50 (8%), HHV-1 (6/50, 12%)	Polz-Gruszka et al. [17]	2015
Poland	Fresh frozen	N.P.	Nested-PCR	EBNA2	12/40 (30%)	-	-	-	-	PCR and L1 gene	6/40 (15%)	28 types of HPV (especially HPV type 16)	6/40 (15%)	BKV (4/40, 10%)	Drop et al. [18]	2017
Iraq	FFPET	From 25 to 78 years old Mean: 53.6 ± 2.11	ISH	EBER	12/30 (40%)	Healthy control (from 23 to 68 years-old; mean: 46.5 ± 2.81)	ISH	EBER	0/15 (0%)	-	-	-	-	-	Ali et al. [27]	2018
						Nodules (from 26 to 74 years old; mean: 44.2 ± 2.14)	ISH	EBER	2/13 (15.4%)	-	-	-	-	-		
						Polyps (from 13 to 71 years old; Mean: 45.7 ± 3.39)	ISH	EBER	1/12 (8.3%)	-	-	-	-	-		
Mexico	FFPET	From 40 to 89 years-old Mean: 64.5	qPCR	LMP2	54/195 (27.7%)	-	-	-	-	PCR and L1 region	93/195 (47.7%)	HPV (especially HPV type 11)	25/54 (46.3%)	MCPV (11/195, 5.6%)	Vazquez-Guillen et al. [3]	2018
						-	-	-	-	PCR and L1 gene	63/195 (34.4%)	HR-HPV (especially HPV type 52)				
Spain	FFPET	From 46 to 80 years-old Mean: 64	ISH	EBER	0/7 (0%)	-	-	-	-	PCR-hybridization and HPV DNA	4/7 (57.14%)	HPV16	-	-	Acuña et al. [19]	2019
Poland	Serum	From 40 to 79 years old	Immunoenzymatic assay	IgG anti-VCA	33/33 (100%)	Healthy patient serum	Immunoenzymatic assay	IgG anti-VCA	20/20 (100%)	-	-	-	-	-	Klatka et al. [6]	2019
				IgM anti-VCA	14/33 (42.42%)	(from 44 to 67 years old)	-	-	-	-	-	-	-	-		
				IgG anti-EBNA1	24/33 (72.72%)		Immunoenzymatic assay	IgG anti-EBNA1	20/20 (100%)	-	-	-	-	-		
Taiwan	FFPET	From 58 to 74 years old Mean: 64	qPCR	BamHI W	22/42 (52%)	Non-malignant lesion (from 37 to 58 years old; mean: 52)	qPCR	BamHI W	8/39 (20%)	-	-	-	-	-	Lee et al. [28]	2021
	Serum		qPCR	BamHI W	2/30 (6.7%)		-	-	-	-	-	-	-	-		
	Serum		Immunofluorescence	IgA anti-VCA	4/30 (13%)		-	-	-	-	-	-	-	-		
	FFPET-tissue microarray		ISH	EBER	28/42 (67%)	Non-malignant lesion	ISH	EBER	22/39 (56%)	-	-	-	-	-		
Netherlands	FFPET-tissue microarray	N.P.	ISH	EBER	0/9 (0%)	-	-	-	-	qPCR HR-HPV	0/9 (0%)	-	-	MCPyV IHC 0/9(0%)	Mulder et al. [20]	2021
							**In vitro study**									
**Country**			**Type of cell**						**Main result**							
UK	Primary cultures of a laryngeal carcinoma					Identification of a 200 kDa protein on the surface of primary cultures of a LC sharing an epitope with the C3d/EBV receptor molecule CD21 of B lymphocytes									Young et al. [11]	1989

N.A., not available; N.P., data extraction not possible, because the authors did not present the age range information of the LC group separately from the other tumors.

**Table 2 pathogens-10-01429-t002:** Results of meta-analysis with random-effects model in carcinoma cases: pooled prevalence of EBV detection and CI at 95% in subgroups analyzed.

Analysis/Subgroup	N *	Pooled Prevalence	95% CI	Heterogeneity Test		
				*Q*	*p*	*I^2^*
**Sample**						
Serum	7	41.66	18.19–65.12	104.99	<0.001	94.3%
Fresh frozen tissue	8	42.30	30.18–54.43	31.22	<0.001	77.6%
Throat washings	1	64.51	47.67–81.35	0.00	–	–
FFPET	12	43.31	27.53–59.10	189.42	<0.001	94.2%
**World region**						
Asia	9	42.37	21.93–62.82	220.81	<0.001	96.4%
Eurasia	3	46.17	34.29–58.05	0.64	0.725	0.0%
Europe	10	48.17	36.87–59.47	30.75	<0.001	70.7%
North America (NA)	5	40.42	22.23–58.60	43.26	<0.001	90.8%
NA and Europe	1	18.18	−4.61–40.97	0.00	–	–
**Development status**						
Developed countries	22	46.37	35.27–57.47	223.87	<0.001	90.6%
Developing countries	6	34.02	19.88–48.17	49.05	<0.001	89.8%
**Overall**	28	43.72	34.35–53.08	347.80	<0.001	92.2%

* N, number of members in each subgroup.

## Data Availability

The data presented in this study are available on request from the corresponding author.

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
