# Peer review of "Epstein-Barr Virus-Associated Carcinoma of the Larynx: A Systematic Review with Meta-Analysis"

_pathogens, 2021, doi:10.3390/pathogens10111429_

Round 1
Reviewer 1 Report
Summary:
Lima et al., sort to conduct a meta-analysis to evaluate the correlation/ associate between EBV and laryngeal carcinoma. To achieve this goal, Lima et al combed two databases searching for keyword EBV and laryngeal carcinoma. Based on their exclusion/ inclusion criteria. Based on the review of these papers, they looked at how (methodology) EBV infection was verified, countries each study was conducted and co-infection with other viruses. They concluded that based on the studies, slightly less than half (43%) of laryngeal carcinoma was associated with EBV and prevalence differs in different control vs LC cases, subgroups and developing vs developed countries. Lima et al., concluded that based on the available studies there is evidence to suggest EBV might be associated with laryngeal carcinoma. Due to the small number of studies looking at the association, additional studies need to be conducted before conclusively defining the role of EBV in laryngeal carcinoma.
Corrections:
Introduction paragraph:
- The first sentence of the introduction (Line 27-30) is a mouthful, either paraphrase or split the sentence to make it more clear.
- The subparagraph on HPV (lines 41-46) fills out of place, why was HPV singled out of all other oncoviruses? Are you gonna be looking specifically at the co-infection? There is no clear justification to the focus on HPV, as your study questions don't mention HPV or co-infections.
Results section:
- In table 2, it would have been great if the age range of the patients was included. It is known that in developing country EBV seroconversion occurs earlier than developed countries. does that play a role in the different prevalence seen between these different group. This would also be important to understand the age range of the healthy control compared to LC cases.
- Assembly error with figure 1, it cuts line 95.
- Figure 1 and table 2 n=28 but the table 1 said n= 33 and abstract said n=31. Its unclear why the number of cases differ within the review.
Discussion:
- Majority of the literature review would have been better suited for the introduction to clearly set up the study.
- Isn't it possible that due to sample size difference between developed and developing countries that's why there is the difference between the presented prevalence?
- In relation to comparison with the control groups "
while the overall analysis showed that EBV-infected individuals are at 2.18 times greater risk of developing laryngeal carcinoma (line 126-127)
"how does this conclusion take into account that prevalence of EBV being 90-95%? - Do you think active (lytic) EBV infection is important for or associated with LC prognosis?
Author Response
Reviewer #1
Corrections:
Introduction paragraph:
- The first sentence of the introduction (Line 27-30) is a mouthful, either paraphrase or split the sentence to make it more clear.
Author Response – The sentence was modified as suggested.
- The subparagraph on HPV (lines 41-46) fills out of place, why was HPV singled out of all other oncoviruses? Are you gonna be looking specifically at the co-infection? There is no clear justification to the focus on HPV, as your study questions don't mention HPV or co-infections.
Author Response – The investigation of co-infections emerged during the analysis of results phase, when we realized that an expressive number of studies reported the occurrence of such events. Regarding the HPV/EBV co-infection, their co-occurrence is well documented in oropharyngeal and anogenital carcinomas, even in breast carcinomas, and we have already addressed a possible cooperation between these two viruses in carcinogenesis in a previous study (de Lima et al., 2019). For this reason, we thought that it would be appropriate to present a brief information about HPV in the introduction. Additionally, as the evaluation of this aspect (co-infections) was not considered during the conceptualization of the research, we kept the research question as originally structured.
Results section:
- In table 2, it would have been great if the age range of the patients was included. It is known that in developing country EBV seroconversion occurs earlier than developed countries. does that play a role in the different prevalence seen between these different group. This would also be important to understand the age range of the healthy control compared to LC cases.
Author Response – Age range data from carcinoma cases were included in a new column in Table 1. We also included age range information in the “sample control” column. About the fact that “in developing countries EBV seroconversion occurs earlier than developed countries”, some considerations were added in this regard in the 3rd paragraph of the topic “3.3. Evidence that corroborate the role of EBV in development of LC”.
- Assembly error with figure 1, it cuts line 95.
Author Response – The position of figure 1 was adjusted as suggested.
- Figure 1 and table 2 n=28 but the table 1 said n= 33 and abstract said n=31. Its unclear why the number of cases differ within the review.
Author Response – The typos were corrected and the quantities were standardized throughout the text. We also included a sentence in the second line of the first paragraph of the topic "2.3. Meta-analysis" to clarify the number of the articles included in the qualitative analysis (systematic review) and in the meta-analysis.
Discussion:
- Majority of the literature review would have been better suited for the introduction to clearly set up the study.
Author Response – We included a paragraph in the introduction in order to denote the importance of the study.
- Isn't it possible that due to sample size difference between developed and developing countries that's why there is the difference between the presented prevalence?
Author Response – Comments in this regard were included in the 3rd paragraph of the topic “3.3. Evidence that corroborate the role of EBV in development of LC”.
- In relation to comparison with the control groups "
while the overall analysis showed that EBV-infected individuals are at 2.18 times greater risk of developing laryngeal carcinoma (line 126-127)
"how does this conclusion take into account that prevalence of EBV being 90-95%?
Author Response – This is a feature that should be considered in all studies exploring the role of EBV in cancers. It is well known that about 90-95% of adults worldwide harbor EBV in circulating B-cells, for this reason there is the possibility of amplification of the viral genome from tumor infiltrating B-lymphocytes when using PCR, thus the importance of histological approaches in this context, to confirm the presence of the virus in malignant cells. We addressed such aspects in the 3rd paragraph of the topic "3.2. Controversial Aspects of the EBV-LC Relationship". In addition, due to the aforementioned motive, we performed the analysis of Figure 1 stratifying the results in subgroups according to the technique employed.
- Do you think active (lytic) EBV infection is important for or associated with LC prognosis?
Author Response – Considering the lack of studies investigating the expression of EBV lytic proteins (or transcripts) in laryngeal carcinomas, it is still premature to draw conclusions about the impact of lytic infection on LC prognosis. However, there is a bunch of evidence that support the role of the lytic infection in the oncogenesis, such as the homology between lytic proteins and human proteins involved in oncogenesis or immunosurveillance (e.g., BHRF1 and Bcl-2; or BCRF1 and IL-10). Also, the lytic protein BZLF1 degrades p53, which probably also contributes to oncogenesis. Another relevant aspect that corroborates the importance of lytic proteins in this process is the “hit-and-hide” mechanism, according to which EBV switches between latent and lytic mode. The hypothesized mechanism would also explain how EBV can contribute to tumor development even in the lytic cycle, and would shed light on the oncogenic effects of a number of lytic EBV proteins. All these aspects were addressed in a previous publication of our team (DOI: 10.1615/CritRevOncog.2020033071).
Reviewer 2 Report
Paper written by Pereira de Lima and co-workers is clair and it is well written and deals with a current and interested topic. However before pubblication need of some modifications.
1-Introduction. I believe that more recent data should be provided to stress the importance of this paper. It is not clear the importance of the study performed, the relapses.
table 2 needs to be edited is difficult to read and understand
Author Response
Reviewer #2
Paper written by Pereira de Lima and co-workers is clair and it is well written and deals with a current and interested topic. However before pubblication need of some modifications.
1-Introduction. I believe that more recent data should be provided to stress the importance of this paper. It is not clear the importance of the study performed, the relapses.
Author Response – We included a paragraph in the introduction in order to denote the importance of the study.
table 2 needs to be edited is difficult to read and understand
Author Response – Table 2 was edited as suggested. Additionally, the first table was wrongly named table 2, the number was corrected.
Reviewer 3 Report
This is an interesting systematic review and meta-analysis about Epstein-Barr virus-associated carcinoma of the larynx.
The paper is well written. However, some issues remain.
The legend of table 1 (wrongly named table 2) refers to 33 studies, but only 31 articles are included in the review. Please correct it.
In table 1 (wrongly named table 2), the authors should better specify the sample type of the control group.
Author Response
Reviewer #3
This is an interesting systematic review and meta-analysis about Epstein-Barr virus-associated carcinoma of the larynx.
The paper is well written. However, some issues remain.
The legend of table 1 (wrongly named table 2) refers to 33 studies, but only 31 articles are included in the review. Please correct it.
Author Response – The number of the first table was corrected. The quantity of studies was also corrected as requested.
In table 1 (wrongly named table 2), the authors should better specify the sample type of the control group.
Author Response – The samples of the control group were better specified in table 1 as suggested.